# The Iodine/Iodide/Starch Supramolecular Complex

**DOI:** 10.3390/molecules29030641

**Published:** 2024-01-30

**Authors:** Szilard Pesek, Radu Silaghi-Dumitrescu

**Affiliations:** Department of Chemistry, Faculty of Chemistry and Chemical Engineering, Babeş-Bolyai University, 11 Arany Janos Street, 400028 Cluj-Napoca, Romania; szilard.pesek@ubbcluj.ro

**Keywords:** amylose, iodine, iodide, starch, UV-vis, DFT

## Abstract

The nature of the blue color in the iodine–starch reaction (or, in most cases, iodine–iodide-starch reaction, i.e., I_2_ as well as I^−^ are typically present) has for decades elicited debate. The intensity of the color suggests a clear charge-transfer nature of the band at ~600 nm, and there is consensus regarding the fact that the hydrophobic interior of the amylose helix is the location where iodine binds. Three types of possible sources of charge transfer have been proposed: (1) chains of neutral I_2_ molecules, (2) chains of poly-iodine anions (complicated by the complex speciation of the I_2_-I^−^ mixture), or (3) mixtures of I_2_ molecules and iodide or polyiodide anions. An extended literature review of the topic is provided here. According to the most recent data, the best candidate for the “blue complex” is an I_2_-I_5_^−^-I_2_ unit, which is expected to occur in a repetitive manner inside the amylose helix.

## 1. Introduction

Starch is a mixture of amylose and amylopectin in a ratio of ~20–30% to 70–80%, both of which are polymers of glucopyranose. Amylose is a linear polymer typically containing 300–3000 (or sometimes much more) monomeric units that are interconnected via α(1→4) glycosidic bonds (cf. Figure 1) [1]. Amylopectin has a similar structure, additionally featuring ramifications via α(1→6) bonds. A small number of ramifications (of the same type as amylose) are in fact also found in amylose. The average number of branches in amylose molecules is size-dependent, ranging from zero to 2–4 branches per molecule. The variations have been generally attributed to differences in chain length, with the observation that most branching occurs at the early stage of amylose synthesis [2]. Both polymers are sparsely soluble in water, with amylose more so than amylopectin. Inside living organisms, the enzyme amylase is mainly responsible for hydrolyzing starch. Dextrins, which are polysaccharides with a low degree of polymerization, are produced by partial hydrolysis of starch; with complete hydrolysis, glucose is formed [3].

Amylose can exist in a disordered amorphous conformation or in two types of helical forms (Figure 2). The first type is a double helix with itself (including forms A and B). The second type is the V form, which consists of a single helix; this is the structure typically discussed in biochemistry textbooks. The V form features an internal cavity that is large enough to accommodate hydrophobic guest molecules such as iodine, fatty acids, or small aromatic compounds. This property has in fact been recently exploited in extensive attempts to design smart drug delivery systems, with amylose/starch serving as the host for inclusion complexes with various molecules of therapeutic potential/use [4,5,6].

In 1930, Katz studied the aging and cooking of bread, and with the help of X-ray diffraction, in the analyzed powder he found, in addition to the type A and B models known until then from native starch, he found another crystalline form of starch, which he called model V. To describe the contribution of this form, he used the German word “Verkleitsterung,” which means gelatinization [7]. He found the same V-type pattern when he prepared pasta, which was precipitated with alcohols [8]. Bear identified other different V-type patterns depending on the precipitation agent used [9]. Native starch was fractionated by Meyer et al., separating amylose from amylopectin using hot water [10].

A- and B-amylose both form parallel-stranded double helices of 6 × 2 glucoses per turn and right-handed [11,12,13,14,15,16] or left-handed [17] turns. Structurally, these two structures differ from each other only in their packing arrangements and water contents. They also differ in biological locations, with A preferentially in grains and B in tubers [11]. A polymorph C of amylose has also been described, consisting of a mixture of A- and B- unit cells [18].

The V form is also found naturally and is structured as a single left-handed helix with six glucose units per turn and a step height of 7.91 to 8.17 Å [11,12,13]. The V form can be isolated through its precipitation from aqueous solution using alcohols, ketones, fatty acids, iodine, or salts that form inclusion complexes (thus behaving similarly to cyclodextrins, α-cyclodextrin, or cyclohexaamylose) [19,20]. The glucose units in V-amylose (as in cyclodextrins) are all in the *syn* orientation. This entails hydrogen bonds between the secondary hydroxyl groups O(3)_n_⋅⋅⋅O(2)_n+l_, as well as O(6)_n_⋅⋅⋅O(2)_n+6_ hydrogen bonds between turns [11,12,13].

Variants of the V-type structure of amylose have been described, each marked with a subscript typically indicating the number of glucose units per shift. The most common variant is the V_6_ form [14], though V_8_ and V_7_ have also been described. The latter would provide even more space for the guest molecule to bind [21]. Depending on the solvent, slightly different crystallographic structures are formed. The crystals grown with water as the solvent, dubbed amylose V_h_, feature six glucose units in the unit cell, aligned as a single strand. Within the helix, there are three water molecules, statistically positioned on six positions. In the spaces between helices, there is one water molecule each, on one of three possible positions. The helices point upwards and downwards in a random fashion [14].

Amylose V is the allomorph known for its deep blue complexes with iodine. The iodine molecules are trapped in the channels within the helices, where molecules of the solvent can also be present. As detailed in the following sections, extensive research has been devoted to understanding the structural features of the amylose–iodine complexes, especially as the blue complexes generally form only when iodide anions are present alongside iodine. This implies the presence of polyiodide species, of which tri-iodide is the most often invoked. Figure 3 illustrates the proposed structures of amylose–I_2_ and amylose–tri-iodide species based on electronic structure calculations. Figure 4 illustrates the representative structures proposed for the iodine/iodide arrangement inside the amylose helix [22].

Cyclodextrins are glucose oligomers featuring the same type of glycosidic bonds as starch, and they can be obtained enzymatically. Depending on their size, some cyclodextrins can arrange themselves in a circular fashion, appearing similar to single turns of the amylose helix, and then forming dimers which generate large cylinders, similar to the amylose helix in its overall structure.

## 2. General Considerations on the Reaction of Starch with Iodine

Colin and Claubry discovered in 1814 the reaction between starch and iodine (or, more specifically, with iodide–iodine solutions, as molecular I_2_ is otherwise very insoluble in water in the absence of I^−^ ions). Since then, the reaction has been found both in organic chemistry classes in school and in qualitative and quantitative analysis courses. Over time, many experiments have shown that the starch–iodine complex shows absorption at ~600 nm: a strong dark blue color [23]. A more detailed description of the reaction was provided by H. H. Landolt in 1886 [24]. The color is mainly due to the complex of iodine with the amylose complex that absorbs at ~620 nm. The affinity of iodine for amylopectin is distinctly smaller (~20 times), and the resulting complex is reddish-violet, with a maximum at ~540 nm. In 1948, Gilbert and Marriott [25] showed that at higher concentrations of iodide, the ratio of iodide ions to iodine molecules increases to at least one, leading to a purple tint in the blue complex.

The nature of the amylose–iodine complex has been debated for many decades, especially in terms of the stoichiometry and charge of the poly-iodine substructures within the helix (see, e.g., Figure 4). It is now generally accepted that iodide ions are also required in this process. Thus, the iodine atoms would align inside the amylose helix as a mixture/combination of I_2_ and I^−^ [21,22,24,25,26,27] with an unusual metallic-like structure [23]. This mixture would not entail isolated I^−^ ions, but rather I_n_^−^ polyiodides (*n* = 3, 5, 9…) [28,29,30,31,32]. The wavelength of the absorbance maximum in the amylose–iodine complex is known to depend on the chain length. Thus, glucose chains of 4 to 6 units yield no color, while those 8 to 12 yield a red color with a peak at 520 nm reminiscent of amylopectin. Longer chains progressively show a bathochromic shift until a length of 30 to 35 units, when the blue color is reached with a peak at 600–620 nm. Very similar spectra are also obtained through the synthetic action of potato phosphorylase on starch when chains of 50–150 units are obtained. 

This relationship between chain length and iodine color has also been applied to branched polysaccharides. Comparing the spectra of the iodine/iodide complexes with dextrin, amylopectin, glycogen, or various synthetic oligosaccharides, estimations of the extent of helical portions available for iodine binding in amylopectin and glycogen were formulated, at 8–18 glucose residues. In line with these observations, hydrolysis of amylose by α-amylase or under acid catalysis gradually changes the absorption maximum from blue to red. On the other hand, hydrolysis with β-amylase leads to a hypochromic but no hypsochromic shift. This can be explained by the fact that β-amylase remains bound to its substrate until it completely degrades it, rather than gradually degrading all the polymer chains at the same time. In this way, little or no red-colored intermediate dextrins remain in the mixture [33]. Variations in color for iodine–amylopectin complexes can be observed due to the difference in structures [34], branched chain length [35], and branching points in amylopectin [36]. If the degree of polymerization in amylopectin entails 15 glucose units, complexation of iodine is not observed at any temperature. If it exceeds 30, then iodine binds to amylopectin at either 1.5 °C or 20 °C [37]. Mould and Synge [38] used potentiometric and spectrophotometric titrations for complexation of iodine/iodide with the products of enzymatic hydrolysis of amylose, i.e., dextrins with different molecular masses. Dextrins of less than 10 glucose units were essentially unreactive, those of 10–25 units were orange, those 25–40 were red, and those of 40–130 were blue. Ono [39] showed that the λ_max_ of the amylose–iodine complex shifted to shorter wavelengths with increasing iodide concentrations; this was interpreted as evidence of the breaking the polyiodine chains by the permeated iodide ions.

## 3. Dependence on the Nature of the Organic (Bio)Polymer

In addition to starch, there is a long list of natural polymers that afford colored complexes with iodine. These include chitosan, glycogen, silk, wool, albumin, cellulose, xylan, and natural rubber. A large number of synthetic polymers have also been described to react with iodine. Examples include poly(vinyl alcohol) (PVA), poly(vinyl pyrrolidone) (PVP), nylons, poly(Schiff base)s, polyaniline, and unsaturated polyhydrocarbons (carbon nanotubes, fullerenes C_60_/C_70_, and polyacetylene) [40]. It is important to note that most of these polymers do not feature helical structures of the type seen in amylose; binding of the iodine on the outside of the organic polymer, or between polymer chains, is likely occurring in such cases.

Differences in the blue color were reported depending on the size (and, implicitly, biological source) of the polymer, be it amylose or related poly and oligo saccharides or other organic polymers [34,36,40,41,42,43,44,45,46,47,48,49,50,51,52,53,54]. Yu et al. [55] showed that when complexation occurs between I_2_/KI and potato amylose, the speciation of iodide varies. Data from Raman and UV-visible spectroscopy were interpreted as evidence that the primary forms of iodine were the monoanions I_3_^−^ and I_5_^−^, but larger units were also present such as I_93_^−^, I_113_^−^, I_133_^−^, and I_153_^−^ (with bands at 460–480, 560–590, 660–700, and 710–740 nm, respectively), with higher iodide concentrations expectedly favoring shorter polyiodide chains. By adding an iodine–potassium iodide solution to a solution of cellulose, Abe [56] obtained an intense blue solution. At 80 °C, the color disappeared, but upon cooling, it reappeared. Takahashi [57] reported a dark purple adduct upon treating chitin with I_2_/KI solution for 24 h at room temperature. Depending on the amount reactant ratios, the iodine content of the adduct was in the range of 9–20%, with one molecular iodine per 6.4 chitin residues. Yajima et al. [58] prepared a purple complex (maximum at 550 nm) by freezing a mixture of chitosan and I_2_/KI solution at −20 °C and then thawing it at 4 °C.

Glycogen, as also discussed above, yields a reddish-brown complex with iodine. The largest maximum is at 395 nm, [59] but as shown by Kumari et al. [60] and Lecker et al. [61], a series of UV absorption bands is also present at 408, 453, 496, 560, 650, and 698 nm. The bands at 408, 453, 560, and 650 nm were comparable to those of amylopectin (412, 458, 550, and 640 nm), and the bands at 496 and 698 nm were assigned to an I_4_ species. The bands at 453 and 560 nm were more visible at higher iodine concentrations.

Stromeyer, in 1815, mentions that wool and silk (i.e., the protein structure therein) give yellow colors on exposure to iodine [37]. Amyloid peptides also yield a blue color with molecular iodine. In the presence of sulfuric acid, this shifts to blue–violet, cf. Aterman [62]. Dzwolak [63] described the formation of an insulin–amyloid complex in the presence of I_2_/KI that is stable up to 90 °C, interpreted as an inclusion complex between the amyloid fibrils and iodine.

Pritchard and Serra [64] reported that reaction of poly(vinyl acetate) (PVAc) with molecular iodine in methanol in the presence of aqueous KI yields a deep-red precipitate (darker at higher iodine concentrations), interpreted as an I_2_-PVAc adduct. Two UV-vis absorbance bands were reported for the complex at 520 and 510 nm. Hughes et al. [65] noted that the red color of the I_2_-PVAc complex is independent of the method through which the polymer is prepared. Solutions of the PVA–iodine complex (I_2_-PVA) feature intense 600–620 nm, 650–680 nm, 480–500 nm, and 350 nm [66] bands depending on the conditions. The complex of I_2_ with poly(vinylpyrrolidone) (povidone or PVP) has a maximum at 361 nm, with the iodine molecule reported to be attached to the PVP matrix through non-covalent interactions with the carbonyl groups. A poly(N-methyl-4-vinyl pyridinium) triiodide was reported to produce a dark-brown powder with molecular iodine in a water–alcohol solution, with bands at 295 nm, 367 nm, and 460 nm [67]. 

Charge transfer complexes of iodine with ferrocenyl-bearing Schiff bases have also been described [68]. Liu et al. [69] showed that the brown poly(ferrocenyl-Schiff) bases turn black after treatment with iodine in acetone. In the infrared (IR) spectra, iodine binding resulted in a decrease of the ~1610 cm^−1^ signal of the Schiff base, accompanied by the appearance of a new wide absorption band at 450–480 nm with a long tail up to ~900 nm in the UV-vis spectra.

Shirakawa [70] examined the reaction of polyacetylene (PAC) with iodine. Strong IR bands were noted at 870 cm^−1^ and 1390 cm^−1^. In the UV-vis spectrum analysis, bands at 365 nm and 502 nm were noted to be suggestive of the presence of I_3_^−^ and I_2_, respectively. A band at 280 nm characteristic of unreacted PAC was noted to decrease in intensity upon doping.

Treatment of poly(p-phenylene vinylene) or phen(p-phenyvinylene) (PPV) with iodine vapor showed a significant effect on its luminescence [71]. 

In principle, due to the differences in reactivity between amylose and amylopectin, the iodine reaction can be used for assessing the amylose content of starch. However, a number of other methods may be more appropriate and accurate, such as polarimetry, anthrone, FTIR, penetrometry, Luff school, gravimetry, X-ray diffraction (XRD), and size exclusion chromatography (SEC) techniques including multi-angle laser light scattering–differential refractive index detection (SEC-MALS-DRI) [72,73,74,75,76,77].

Three bands in the UV-visible–NIR spectra, at 688 nm, 1724 nm, and 2292 nm, were interpreted as evidence for a PPV–iodine complex. Polybutadiene, poly(cis-isoprene), and their copolymers were reported to display an intense absorption band at 305 nm in the UV spectrum (with some variations depending on the solvent) upon treatment with iodine [78]. Sreeja et al. [79] found that acrylonitrile butadiene rubber developed two new broad bands appearing between 300 nm and 500 nm, while the 262 nm band attributed to the C=C bonds decreased. Poly(β-pinene) treated with iodine vapor was shown by Vippa et al. [80] to produce bands at 310 nm and 400 nm in the UV-vis spectrum, with the latter interpreted as evidence for a charge transfer between the double bond and I_2_.

## 4. Geometrical Data

Crystalline units in starch and in its partial degradation products have long been explored [81,82,83]. Transmission electron microscopy [84], light microscopy [85], the use of atomic force microscopy (AFM) [86,87,88,89,90,91,92], and X-ray and neutron small angle scattering [93] have allowed for some partial visualization of repeating units in amylose. Microstructures were observed, which were called nodules, initially with diameter variations between 150 and 300 nm [84], and also smaller particles of 20–50 nm [85,87,88,93] (e.g., 130–250 nm for pea starch [92], 20–50 nm for potato starch [90], or 10–30 nm for corn granules [84]). Using combined methods of X-ray diffraction and stereochemical packing analysis of the amylose–iodine complex, Bluhm and Zugenmaier [36] reported that the iodine atoms aligned almost linearly in the center of the amylose chain. Eight hydration water molecules were found per unit cell, located in the amylose helices. The amylose left-handed helix was reported to have an outer diameter of~13 Å and a pitch of 8 Å (six 1,4-glucose units per pitch), hosting an internal cavity of ~5 Å [28,30,31,32,34,36,94,95,96,97]. Some statistical disorder was noted within the polyiodide chain, with an average iodine–iodine distance of ~3.1 Å. This value is larger than the 2.67 Å in molecular I_2_ as well as than the 2.90 Å in ionic I_3_^−^, but it is distinctly shorter than the 4.3 Å sum of van der Waals radii for two iodine atoms. With the help of the electron–gas theory, the 620 nm maximum in the UV-vis spectrum was suggested to be due to 14 iodine atoms when assuming an equidistant distance of 3.1 Å. [98,99] However, the X-ray diffraction data so far cannot distinguish between a structure where the iodine atoms are placed equidistantly at 3.1 Å vs. a structure consisting of I_2_ and/or I_n_^−^ units (with internal I-I distances lower than 3.1 Å) placed in non-covalent contact with each other. For instance, I_n_^−^ units with *n* = 5–7 (and assuming the same internal I-I distance as in I_3_^−^) placed at 4.3 Å from each other would allow an average distance of 3.1 Å across the crystal structure, in line with experiments. If assuming the intermolecular distance to be shorter than the sum of van der Waals radii (as expected in complexes displaying charge-transfer bands), the experimentally observed iodine–iodine distance of 3.1 Å could be reached with, e.g., I_3_^−^ units placed at 3.7 Å from each other or I_5_^−^ units placed at 4.1 Å. The same average distance could be obtained by a chain of I_2_ molecules placed within 3.6 Å of each other.

## 5. The I_2_-Only Hypothesis

Many textbooks still list the amylose structure as featuring I_2_ units aligned inside the helical channel. The driving force for this arrangement would be the hydrophobic character of the I_2_ molecules, which thus escape the solvent to align in a more hydrophobic environment inside the helix [100]. The non-covalent interactions between I_2_ would then facilitate charge-transfer bands to appear, resulting in the intense blue color. NMR and UV-vis studies have shown that I^−^ ions are not involved in the iodine–amylose helix, but they help to dissolve iodine in water [55]. Simulations of the UV-vis spectra using semiempirical INDO configuration interactions have been reported to support a (C_6_H_10_O_5_)_16.5_···(I_2_)_3_ stoichiometry for the amylose–iodine complex, refuting instead I_n_^−^ (*n* = 3, 5, or 7) as possible candidates [101].

Water molecules were reported to modulate the structure of the iodine–amylose complex [102]. It was also noted that neither the dimensions of the amylose helix nor the rigidity or the helical vs. random coil secondary structure within the amylose polymer were affected by iodine binding—all of which suggest no strong specific inter-molecular bonding between amylose and iodine. This is also consistent with a simple inclusion complex involving neutral molecules. In contrast, the charge in I_n_^−^ may have been expected to induce local changes in the neighboring polyscachharide units [103,104,105,106,107,108,109,110].

According to semiempirical calculations, the binding of water or of interspersed water and iodine molecules inside the amylose helix results in slight steric distortions of the polymer [22]. The water molecules were found to adhere closely to the walls of the internal channel of amylose and to exclude I_2_ from interactions, suggesting that I_2_ molecules alone would be unable to dislocate water molecules from inside the amylose helix. In fact, I_2_ solutions (in alcohol, so as to not require iodide for solubilization) were only found to be effective in binding to starch at high temperatures, while at lower temperatures, vapor I_2_ easily adsorbs/binds to solid/dry amylose in the absence of water [111].

## 6. Poly-Iodine Anions as Candidates

It is in fact now generally accepted that the formation of the intense blue color in the amylose–iodine reaction requires iodide ions not only simply as an accessory that allows for the solubilization of I_2_ in water, but also because a combination of I_2_ and I^−^ chains is present inside the amylose helix, most likely involving I_n_^−^ units [30,31,32,33,34,35,36]. The (poly) anionic character of these guest ligands may be taken, as discussed above, to be at odds with the hydrophobic nature of the interior cavity of the helix, especially as no counter ions have been discussed or are presumed to be present throughout the cavity. This issue has been addressed by proposing a structure consisting of alternating tri-iodide units and I_2_ molecules, where the more hydrophilic ends/entrances of the helix would remain unoccupied by iodine [112]. To explain the fact that at room temperature, molecular iodine can bind to amylose in the solid state but not in solution, it was noted that the secondary structure of amylose, including the internal diameter of the helix, varies depending on the environment (e.g., solvent, ionic strength, pH, temperature, surfactants) [49,50,54,113,114,115,116,117]. Moreover, the solid-state amylose–I_2_ complex is not stable in water [118]. To complicate matters, it has also been reported that in the iodine–amylose complex (somewhat similarly to the above-discussed cases of other organic polymers), large amounts of iodine can associate outside the helical cavity, with inter- and intra-chain associations also important [29]. Theoretical studies of the complex have been interpreted as evidence that at low temperatures, an I_6_ structure dominates, while at higher temperatures, nonlinear geometries also appear. However, the experimental data have led to descriptions of the amylose-bound iodine chains as featuring 3–4 to 14–15 atoms, and as high as 160 atoms [31,32,33,34,35,36]. Potentiometric titrations at low iodide concentrations have been interpreted as evidence for a 3/2 I_2_/I^−^ ratio (hence, formally I_8_^2−^). However, as the structure appears to be consistently affected by the concentrations of the reactants (especially the iodine–iodide ratios, as followed, e.g., by UV-vis and circular dichroism titrations), I_4_^−^, I_7_^−^, I_9_^−^, I_6_^2−^, I_8_^2−^, I_10_^2−^, I_4_^2−^, I_6_^−^, and I_24_^2−^ structures have also been proposed [29,30,32,50,52,53,119,120,121,122]. Rawlings and Schneider [123], using the statistics of binding isotherms of I_2_/I_3_^−^ and the large variation of the value of the term of the ratio (R), determined the intrinsic binding constant of I_3_^−^ to amylose, whose value was much higher than that of I_2_. Likewise, the mixed binding energy between I_2_-I_3_^−^ exceeded those between I_2_-I_2_ and I_3_^−^-I_3_^−^ species [29]. Statements [97] which assume that I^−^ is not required sometimes do not take into account that it is formed through the hydrolysis of I_2_ itself. Also, it is known that by adding an acid to an aqueous solution of iodine, the blue color is suppressed [26,124].
I_2_ + H_2_O = I^−^ + H^+^ + HOI
3I_2_ + 3H_2_O = IO_3_^−^ + 5^−^+ 6H^+^
I_2_ + H_2_O = H_2_OI+ + I^−^

In addition to classical short units such as I_3_^−^ ions and iodine molecules, the presence of I_5_^−^ [125] and I_7_^−^ [126] ions was also suggested. These proposals can be based on the fact that the composition of the unit is variable and influenced by the degree of polymerization of the amylose and the concentration of the iodide ion. At the end of the amylose chain, there are 7–8 glucose residues, which do not participate in iodine binding [118].

Stopped-flow UV-vis and circular dichroism (CD) kinetics have revealed that shorter chains of iodine enter the amylose helix very fast (less than 1 millisecond) and then rearrange rapidly inside the helix without further contributions from the excess iodine/iodide in the solution [28,127]. The optical rotatory dispersion (ORD) spectrum of the amylose–iodine complex was noted to change even when no changes in the UV-vis spectrum were observed; this was interpreted as evidence for complex dynamics of the helix during which the length of the poly-iodine chains remained unaffected [128]. In the structure of the synthetic complex (benzamide)_z_ H^+^I_3_^−^, a poly-I_3_ structure was reported, and a range of kinetic, spectroscopic (UV-vis, CD, Raman, X-ray absorption), and thermodynamic data have also been interpreted to support such a structure in the amylose–iodine complex [122,129,130,131]. On the other hand, the X-ray diffraction data on the amylose–iodine complex have been interpreted to be inconsistent with arrangements consisting of only single I_2_ or only I_3_^−^ [35,132,133]. An I_5_^−^ structure was proposed for the amylose–iodine complex based on the similarities between the Raman and Mössbauer spectra and those of polycrystalline (trimesic acid···H_2_O)_10_H^+^I_5_^−^ [129,134]. Of the main bands in the Raman spectra of starch–iodine complexes, at 27, 55, 109, and 160 cm^−1^, three were assigned to I_5_^−^ as the dominant species, while the 109 cm^−1^ band was assigned to I_3_^−^ being present as a minor species or impurity [135,136,137,138,139]. Further resonance Raman, ^129^I Mössbauer spectroscopy [129,140] and X-ray diffraction [132,141,142] data have supported I_5_^−^ or I_2_·I_3_^−^ as the dominant species in the amylose–iodine complex. Equilibrium studies in solution have confirmed that I_5_^−^ is present as a free species, with I_4_^2−^ and I_6_^2−^ also being present at higher I_2_ or I^−^ concentrations [143,144,145]. Teitelbaum et al [125] presented evidence that the I_5_^−^ ion is present in the helix. Part of this explanation may be that the formation of the I_5_^−^ ion in the solid complex is produced by the hydrolysis or alcoholysis of iodine, and the amylose studied in this case was freed from water and alcohols [118,125]. Analyzing the absorption spectra of an aqueous solution of iodine at pH = 4.8 in acetate buffer, three peaks were found at 286, 350, and 460 nm [146]. The peaks at 286 and 350 nm were attributed to the presence of I_3_^−^ ions. If amylose was added to this solution, the peak at 286 nm disappeared, the peak at 350 nm had a slight enhancement, and the peak at 460 nm shifted to 620–650 nm. By adding iodic acid to amylose, the blue color does not appear; this indicates the necessity of the presence of anions for the formation of the colored complex [26,124,147]. 

## 7. Structural Role of the Solvent

Benesi and Hildebrand [148] in 1949 showed that iodine is pink–red in benzene, purple in CCl_4_ (similarly to vapor iodine), and reddish-brown in alcohol—all of which were interpreted as evidence for charge transfer complexes. Bernal-Uruchurtu et al. [149] provided a microscopic explanation of the interaction between iodine and water. Kereev and Shnyrev studied iodine, iodide [150], and triiodide in water and noted a band at 203 nm, which they attributed to iodine, and a band at 461 nm, which was then assigned to a H_2_O-I_2_ charge–transfer complex. To complicate such studies, I_3_^−^ is formed by the oxidation of water with iodine. Analyzing iodine solution in water, the absorption bands of I_3_^−^ are at 284 nm and 351 nm. Density functional theory (DFT) calculations showed that two I_2_ molecules can bind to two O lone pairs in H_2_O in a nearly tetrahedral geometry with a stabilization energy of 8.3 kcal/mol. This complex was described as involving a charge-transfer character, with a band in the UV-vis spectrum due to the excitation of electrons from the lone pair of oxygen of the molecular complex at 202 nm, where the molecular orbitals of iodine are destabilized and those of water are stabilized. When iodine is bound to water, the lone pairs on oxygen become equivalent due to hybridization [151]. A small effect of water molecules on the structure of the iodine–amylose complex was described, as previously discussed [102].

## 8. The I_5_^−^-I_2_ Hypothesis

Recent DFT calculations on iodine/iodide chains have been interpreted as evidence that without iodide, the blue color cannot be formed in the starch–iodine system [22]. These simulations propose that the nature of the complex consists of alternating sets of I_2_ and I_x_^−^ units, where the nature of the charge transfer bands responsible for the blue color involves transfer from the I_x_^−^ σ* orbitals (HOMO) to the I_2_ σ* LUMO orbitals (cf. Figure 5). By analyzing the TD-DFT-computed (time-dependent density functional) UV-vis spectra of various candidates (I_2_ chains vs. mixtures of I_2_ and I_x_^−^ with various values of x) and cross-checking with DFT geometry optimizations, a unit of I_2_-I_5_^−^-I_2_, in a repetitive manner within the amylose helix was the only structure that would fit the experimental data. [22] Poly-I_2_ structures were shown to be responsible for the enhanced blue color under certain conditions (e.g., consistent with the experimental observations on dry/solid amylose). Based on semiempirical calculations, poly-I_n_^−^ structures were found to be unlikely to exist inside the amylose helix (as no distinct local energy minima were identified for such arrangements). Moreover, TD-DFT simulations of the UV-vis spectra of such chains were found to be less consistent with the experiments compared to I_2_/I_n_^−^ pairs. Charge transfer bands from the occupied I_n_^−^ (*n* > 3) σ* to the empty I_2_ σ* orbital were instead found to be reasonably responsible for the blue color. Of these, the I_2_-I_5_^−^-I_2_ trimeric assemblies (i.e., *n* = 5) were the smallest units that represented the local minima in DFT geometry optimizations. These DFT-optimized units remarkably showed average iodine–iodine distances essentially identical to the 3.1 Å value seen experimentally in the iodine-amylose complex. The distinct charge-transfer character of the UV-vis bands (cf. Figure 5) also brings about a strong dependence on the dielectric constant in the region ε~1–30, which in turn was proposed to explain at least part of the dependence of the UV-vis properties of the amylose–iodine/iodine complexes on various external factors that may subtly affect amylose architecture and hence exposure of the interior cavity to solvent (e.g., temperature, other solutes, solvents, chain length) [22].

## 9. Conclusions

The iodine–starch (or, more specifically, iodine–amylose) reaction has a two-century history and a wide range of practical applications. Similar reactions occur with other organic polymers. The nature of the reaction is generally accepted to entail the alignment of iodine atoms inside the amylose helix. However, the structural details are still a source of confusion in many current reference sources, with alternative explanations given such as poly-I_2_ (chain of neutral iodine molecules), poly-I_3_^−^ (chain of I_3_ anions), poly-I_x_ (chains of anionic structures of various lengths), or mixtures of I_2_ and I_x_^−^. The most recent data suggest that the best explanation is a (probably repetitive) I_2_-I_5_^−^-I_2_ unit.

## Figures and Tables

**Figure 1 molecules-29-00641-f001:**
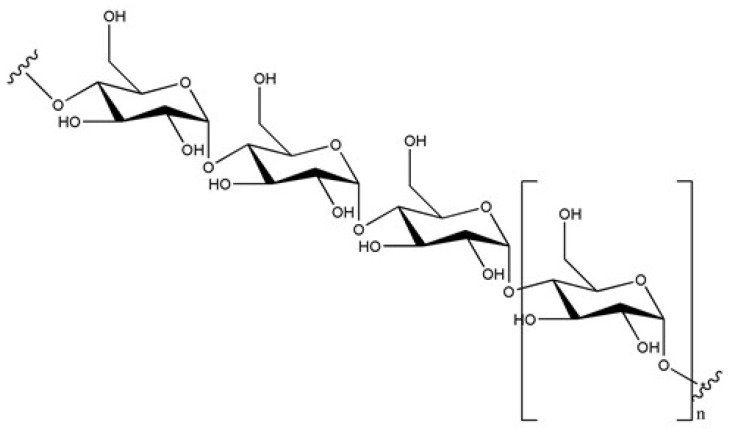
Molecular structure of amylose.

**Figure 2 molecules-29-00641-f002:**
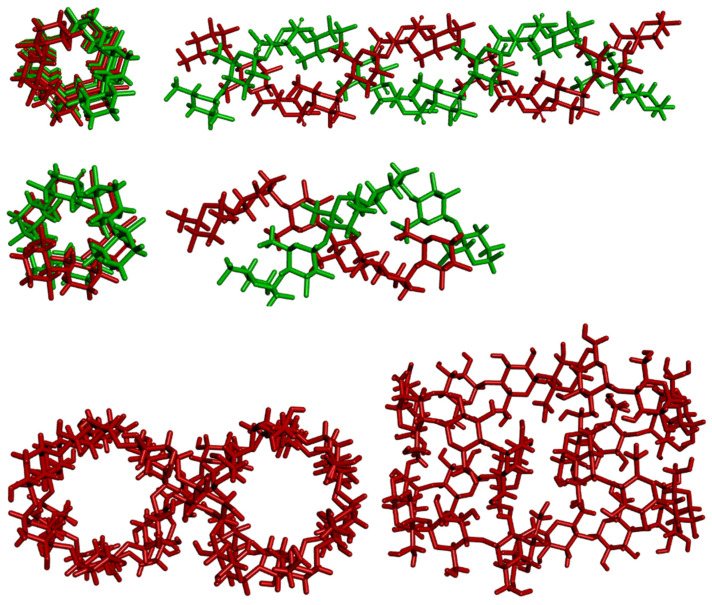
Molecular models of helices of A-type, B-type, and V-type cycloamylose.

**Figure 3 molecules-29-00641-f003:**
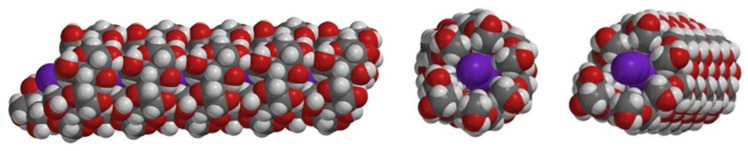
Structures of AM1-optimized amylose models: A-I_2_ and A-I_3_^−^ [22].

**Figure 4 molecules-29-00641-f004:**
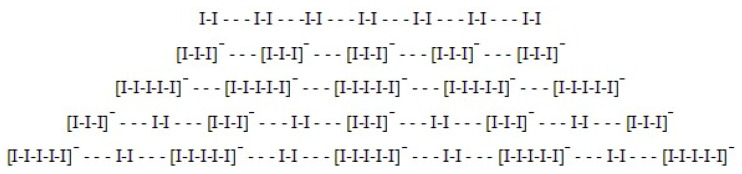
Structures of I_2_, I_3_^−^, I_5_^−^, I_3_^−^-I_2_-I_3_^−^, I_5_^−^-I_2_-I_5_^−^, proposed as possible sources of the blue color in the iodine–iodide–amylose complexes.

**Figure 5 molecules-29-00641-f005:**
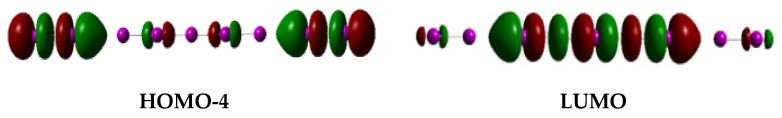
Molecular orbitals in the I_2_-I_5_^−^-I_2_ that are proposed to be responsible for the ~600 nm band in the iodine/iodide–amylose complex. Iodine atoms are shown in violet, HOMO/LUMO lobes are shown in red/green [22].

## Data Availability

Not applicable.

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
