# Peer review of "The Iodine/Iodide/Starch Supramolecular Complex"

_molecules, 2024, doi:10.3390/molecules29030641_

Round 1
Reviewer 1 Report
Comments and Suggestions for Authors
Although the formation of starch-iodine complexes has been known for a long time and used both for scientific and didactic purposes, the authors showed new aspects of the issue. It is interesting that despite such a long history of application of this reaction, there are still doubts about the nature of the starch-iodine-iodide interaction. Although the review is not very extensive, the authors cited quite a lot of literature references.
Please find my comments in the pdf file.

The English language requires minor corrections for easier understanding of the presented issues.
Author Response
To some extent, we agree with the Reviewer. We believe that by following the descriptions in the continuation of the review, we can get a broad overview of the structure of starch and the reactions of starch with iodine. Some changes were also performed with this in mind.
Reviewer 2 Report
Comments and Suggestions for Authors
1. How it will help to increase the solubility of drug.
2. Give the Table for the different formulations reported alone or in combination with these carrier.
3. How this combination will be better than other carrier system.
4. Is there any preclinical data reported for these.
5. Its application to delivery system is missing.
Author Response
We have found no animal data on the starch-iodine/iodide system or delivery system applications. However, some proof-of-concept laboratory experiments are available and have been added to the manuscript.
Reviewer 3 Report
Comments and Suggestions for Authors
1 Amylose also has limited number of branches.
2 Generally, for a review paper, the cited references for one thing should not only be one such as in Line 91, Line 53, 3~5 references are recommend as more evidences are needed, please revise this in this manuscript.
3 Line 104-107, what do you mean here? And any references?
4 Line 108, Line 128, Line 338, Please add references?
5 L310, delete the repeated “of”
6 L315, delete the repeated “data”
7 More graphs are needed for a review paper to make the manuscript more meaningful.
8 As it has been reported that using iodine method to measure the amylose content of starch was not as accurate as other methodologies (such as SEC-DRI), especially for low amylose content starches, could the author in the manuscript also review the application of the theory?
Author Response
- Amylose also has limited number of branches.
Reply: We agree with Reviewer 3, there are some situations where amylose can have some ramifications, we have corrected this aspect in the manuscript.
- Generally, for a review paper, the cited references for one thing should not only be one such as in Line 91, Line 53, 3~5 references are recommend as more evidences are needed, please revise this in this manuscript.
Reply: On Line 91 we deleted the reference, all references appear at the end of the sentence, on line 95. For Line 53 we have 3 references, which appear in line 56.[please note: original numbering used in this reply]
- Line 104-107, what do you mean here? And any references?
Reply: In Lines 104-107 there is a reference to the end of the phrase, Line 112.
- Line 108, Line 128, Line 338, Please add references?
Reply: That phrase also belongs to Line 108, with reference to Line 112.Line 128 has reference to Line 131, and Line 338 has reference to Line 342.
- L310, delete the repeated “of”
Reply: Of is repeated in Line 309, but we believe that both are needed to understand the content.
- L315, delete the repeated “data”
Reply: We do not use ”data” in Line 315, it appears once in Line 312.
- More graphs are needed for a review paper to make the manuscript more meaningful.
Reply: As per Reviewer 3’s suggestion, we have added graphs to review that can make the manuscript more meaningful, easier to navigate, and easier to understand.
- As it has been reported that using iodine method to measure the amylose content of starch was not as accurate as other methodologies (such as SEC-DRI), especially for low amylose content starches, could the author in the manuscript also review the application of the theory?
Reply: Indeed, the reaction with iodine is by no means the most suitable for determining the amylose content of starch, better than the one proposed by the reviewer, but besides these there are other suitable methods in this sense. Methods that can be used to analyze the starch content include the methods of polarimetry, anthrone, FTIR, spectrophotometry, penetrometry, and Luff schoorl. Amylose content can be analyzed by using the spectrophotometry and gravimetry. While the methods that can be used to analyze the amylopectin content of a material, including the X-Ray diffraction (XRD), spectrophotometry, and gravimetry